# Multimodal Procedural Planning via Dual Text-Image Prompting

**Task:** How to make traditional szechuan chicken?

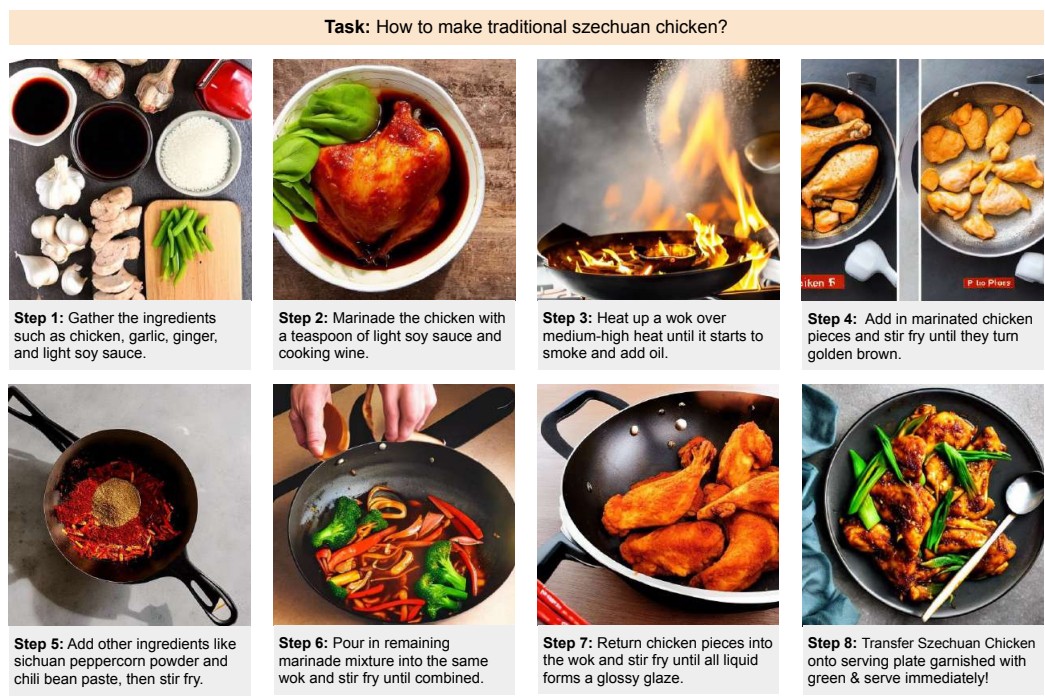

**Step 1:** Gather the ingredients such as chicken, garlic, ginger, and light soy sauce.

**Step 2:** Marinade the chicken with a teaspoon of light soy sauce and cooking wine.

**Step 3:** Heat up a wok over medium-high heat until it starts to smoke and add oil.

**Step 4:** Add in marinated chicken pieces and stir fry until they turn golden brown.

**Step 5:** Add other ingredients like sichuan peppercorn powder and chili bean paste, then stir fry.

**Step 6:** Pour in remaining marinade mixture into the same wok and stir fry until combined.

**Step 7:** Return chicken pieces into the wok and stir fry until all liquid forms a glossy glaze.

**Step 8:** Transfer Szechuan Chicken onto serving plate garnished with green & serve immediately!

Figure 1: Our dual Text-Image Prompting (TIP) model generates coherent and authentic *multimodal procedural plans* towards a high-level goal, providing useful guidelines in task completion.

## ABSTRACT

Embodied agents have achieved prominent performance in following human instructions to complete tasks. However, the potential of providing instructions informed by texts and images to assist humans in completing tasks remains underexplored. To uncover this capability, we present the multimodal procedural planning (MPP) task, in which models are given a high-level goal and generate plans of paired text-image steps, providing more complementary and informative guidance than unimodal plans. The key challenges of MPP are to ensure the informativeness, temporal coherence, and accuracy of plans across modalities. To tackle this, we propose Text-Image Prompting (TIP), a dual-modality prompting method that jointly leverages zero-shot reasoning ability in large language models (LLMs) and compelling text-to-image generation ability from diffusion-based models. TIP improves the interaction in the dual modalities using Text-to-Image Bridge and Image-to-Text Bridge, allowing LLMs to guide the textual-grounded image plan generation and leveraging the descriptions of image plans to ground the textual plan reversely. To address the lack of relevant datasets, we collect WIKIPLAN and RECIPEPLAN as a testbed for MPP. Our results show compelling human preferences and automatic scores against unimodal and multimodal baselines on

WIKIPLAN and RECIPEPLAN in terms of informativeness, temporal coherence, and plan accuracy. [1]

# 1 INTRODUCTION

Recent advances in embodied (Huang et al., 2022; Anderson et al., 2018) and conversational Qiu et al. (2021) agents achieve prominent performance in task completion as humans by following instructions informed by texts and images. However, to what extent the models can provide useful guidelines for humans to complete the task remains underexplored. To uncover this, we propose the multimodal procedural planning task (as shown Figure 1). The task aims to generate goal-conditioned (e.g. "How to make traditional szechuan chicken") text (e.g. "a teaspoon of light soy sauce" explain how to marinade chicken in Step 2) and image (e.g. help identify the ingredients "chicken, garlic, ginger, and light soy sauce" in Step 1) plans as useful guidelines to assist humans in task completion.

Previous work (Huang et al., 2022) explored the generation of procedural plans in text-only form. In contrast, we generate both text and image plans, which provide guidance to perform tasks that acquire complementary information from multimodal contexts. Generating plans in both text and image form poses new challenges since the generated plans should: a) be *informative* enough in both the text and image modalities, b) obey commonsense temporal *coherence*, such as the order of steps, and c) achieve high plan *accuracy*, indicating the complementary and alignment among multimodal plans.

Despite significant progress (Kojima et al., 2022; Song et al., 2022) in the development of large language models (LLMs), they are unable to generate images. Existing text-to-image (T2I) models can generate high-quality images conditioned on textual instructions (Ramesh et al., 2022; Rombach et al., 2022; Brooks et al., 2022). However, they are limited in their ability to generate images that require complex text comprehension, such as temporal reasoning (e.g. "learn basic surf safety *before* hitting the waves") and physical reasoning (e.g. "*pick up* the wine glass"). Additionally, generating text and image plans separately using LLMs and T2I models results in inconsistency and incoherence between the two modalities.

In this paper, we propose Text-Image Prompting (TIP), a novel dual-modality prompting framework that jointly leverages the capabilities of LLMs and T2I models for multimodal procedural planning. We first generate vanilla text plans by querying LLMs (Kojima et al., 2022) for step-by-step procedures. To generate textual-grounded image plans, we devise the Text-to-Image Bridge (T2I-B), which elicits the complex language comprehension abilities of LLMs to assist T2I models in generating informative and text-aligned image plans. Similarly, we generate visual-grounded text plans using the Image-to-Text Bridge (T2I-B), which verbalizes the image plans and injects them back into LLMs to assist the text plans revision, thereby improving informativeness. The temporal coherence of the generated plans is improved considering the context of both text and image. Benefiting from our dual-modality prompting, our generated plans are complementary and aligned across text and image modalities.

To address the lack of suitable datasets for evaluating multimodal procedural planning, we collect the WIKIPLAN and RECIPEPLAN datasets for benchmarking the task. We empirically evaluate the effectiveness of TIP on WIKIPLAN and RECIPEPLAN in a zero-shot setting and compare it with various baselines. Our results demonstrate that TIP generate plausible multimodal plans that are informative, temporally coherent, and accurate. Our work highlights the potential of combining knowledge from LLMs and T2I models to uncover multimodal zero-shot planning capabilities. Our main contributions are as follows:

- We introduce the multimodal procedural planning task and evaluate model performance using our collected WIKIPLAN and RECIPEPLAN datasets.
- We propose Text-Image Prompting (TIP), a dual-modality prompting approach that elicits procedural knowledge jointly from LLMs and T2I models, enabling visual-grounded text plans and textual-grounded image plans.
- We show that TIP substantially improves performance in terms of textual and visual informativeness, temporal coherence, and plan accuracy on human and automatic evaluations.

---

[1] Our code and data are provided in supplemental materials.

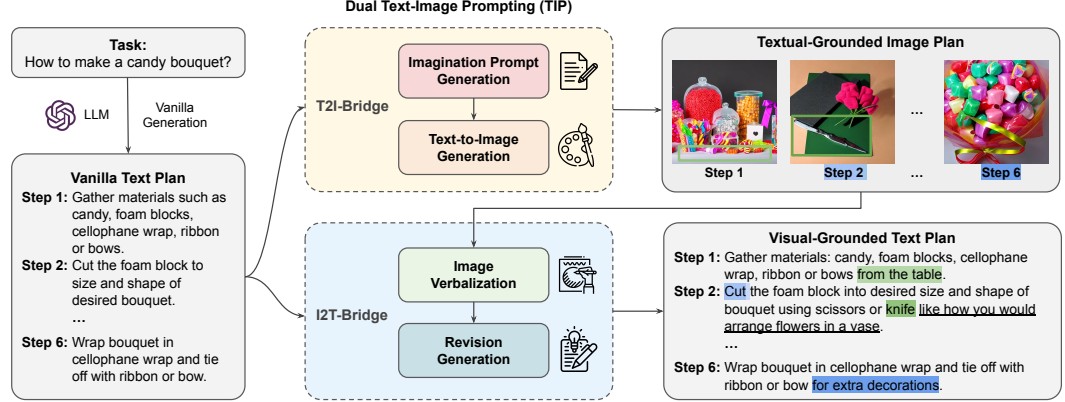

Figure 2: The vanilla text plan is generated using LLM. Our Text-Image Prompting (TIP) generates the textual-grounded image plan using T2I-Bridge (Fig. 3) and the visual-grounded text plan using I2T-Bridge (Fig. 5). The colors blue and green highlight the improved grounding in text and image respectively.

## 2 RELATED WORK

**Procedural Planning**   This task (Zhang et al., 2020; Chang et al., 2020) has gain much attention in various aspects, including robotics Tellex et al. (2011); Jansen (2020); Brohan et al. (2022), vision-and-language navigation Anderson et al. (2018), conversational assistants Ilievski et al. (2018); Qiu et al. (2021; 2022); Yang et al. (2022a), and animation Zhao et al. (2022). Recent work is extended to the multimodal scenarios (Wu et al., 2022; Song et al., 2022; Wang et al., 2022c). In this work, we explore the multimodal procedural planning that generates goal-conditioned text and image sequences grounded in a multimodal context.

**Multimodal Generative Models**   Recently advanced diffusion models Ramesh et al. (2022); Rombach et al. (2022) have shown remarkable abilities in generating high-quality images given text prompts. However, generating images with desired semantics requires proper prompts, which often come from a number of trials and errors Liu & Chilton (2022). To get more controllable generations, researchers have used large language models (LLMs) to expand input prompts with rich contextual knowledge. InstructPix2Pix Brooks et al. (2022) combines the knowledge of GPT-3 and Stable Diffusion to generate large-scale examples of image editing as training data. In turn, recent advances in large-scale models based on transformers (Li et al., 2022; Wang et al., 2022a) exhibit incredible ability in image captioning, describing the given image using natural language.

**Injecting Visual Knowledge in LLMs**   Incorporating visual knowledge into large language models through visual imagination is a promising area of research. This can be achieved through the use of existing images as augmented visual features for language models, or through the generation of images to provide additional visual supervision to language models (Yang et al., 2022b). Studies such as Zhang et al. (2021b); Yang et al. (2022b); Zhu et al. (2022a); Lu et al. (2022b); Liu et al. (2022) have demonstrated the effectiveness of this approach. Our proposed TIP exploits the image descriptions in language form to inject the visual knowledge into LLMs and elicit its potential zero-shot reasoning ability to ground the textual sentences in the verbalized visual context.

## 3 OUR APPROACH

### 3.1 PROBLEM DEFINITION

We formulate multimodal procedural planning as a conditional text and image sequence generation problem. Given a high-level goal $\mathcal{G}$ in natural language form, the model generates a sequence of low-level steps $\mathcal{S} = \{s_1, s_2, ..., s_n\}$. Each step $s_i$ in the sequence is represented by a paired text $t_i$ and image $v_i$ at timestep $i$. The text plan $\{t_1, t_2, ..., t_n\}$ and image plan $\{v_1, v_2, ..., v_n\}$ are both intended to be informative in their respective modalities and complementary across modalities. The

final multimodal procedural plans ($\mathcal{S}$) is the combination of the text plan and image plans, which describe the procedure of completing the high-level goal.

## 3.2 METHOD OVERVIEW

We first elicit the zero-shot step-by-step reasoning ability in large language models (LLMs) to generate a vanilla text-only plan (left part in Figure 2). To enable grounding in multimodal context, we propose Text-Image Prompting (TIP), a dual-modality prompting method (middle part in Figure 2) upon LLMs and multimodal generative models: (1) Text-to-Image Bridge (T2I-B): we generate the visual imaginative prompt that translates the complex textual instructions (vanilla plan in Figure 3) into explicit scene descriptions (prompt in Figure 3) for text-to-image models. (2) Image-to-Text Bridge (I2T-B): we verbalize the image plan with the image captioning model for generating prompts (red highlighted template in Figure 5) that elicit the revision ability of LLMs with awareness of context. Figure 2 depicts how TIP implements multimodal procedural planning by connecting LLMs and multimodal generative models (Image Caption Model, Text-to-Image Model) with our T2I-B and I2T-B, grounding the image plansin textual context and the text plan in visual context respectively (right part in Figure 2).

## 3.3 VANILLA TEXT PLAN GENERATION

We first elicit procedural knowledge of LLM to generate vanilla text plan using Zero-shot Chain-of-Thought (Kojima et al., 2022) that does not require heavy human-engineered few-shot examples. Specifically, we leverage InstructGPT (Ouyang et al., 2022) to generate a goal-conditioned step-by-step procedure with the template "*[TEMPLATE]* Task: *[GOAL]*?". *[TEMPLATE]* represents the hand-crafted template to extract the procedural knowledge from LLM. We extend the template "Let's think step by step" (proposed in (Kojima et al., 2022)) as "What's the step-by-step procedure of" for procedural planning. Then we replace the input slot *[GOAL]* with the given task name $T$ (the high-level goal description) as the prompt $P$ to be fed into the LLM. The LLM then outputs goal-conditioned subsequent steps $\mathcal{W} = \{t_1, t_2, ..., t_n\}$ using greedy decoding as our initial textual plan, which is conditioned only on the task name $T$ in zero-shot generation manner.

## 3.4 TEXTUAL-GROUNDED IMAGE PLAN GENERATION WITH TEXT-TO-IMAGE BRIDGE

Our Text-to-Image Bridge (T2I-B) in Figure 3 leverages LLM to bridge the gap between the language understanding capabilities of LLM and the ability of language-conditioned image generation in the text-to-image model. T2I-B elicits visual imagination in LLM to generate explicit scene description (imagination prompt) for text-to-image model conditioned on the vanilla plan.

**Imagination Prompt Generation** We encourage LLM to revise the prompt that already processes the physical or temporal meaning residing in the original textual plan. To access this, for each step, we use the prompt $P_{t2i}$ "*[STEP] [T2I-B]*" that concatenates the original generated textual plan at step $i$ and the Text-to-Image Bridge template. *[STEP]* represents one of the subsequent steps generated from LLMs. For *[T2I-B]*, we use the trigger sentence similar to "What do I need to draw in the picture to describe the above text?". With this Text-to-Image Bridge guided prompt $P_{t2i}$, the text-to-image model then generates the textual grounded image at each timestep to compose the final sequence of visual plan $\mathcal{V} = \{v_1, v_2, ..., v_n\}$.

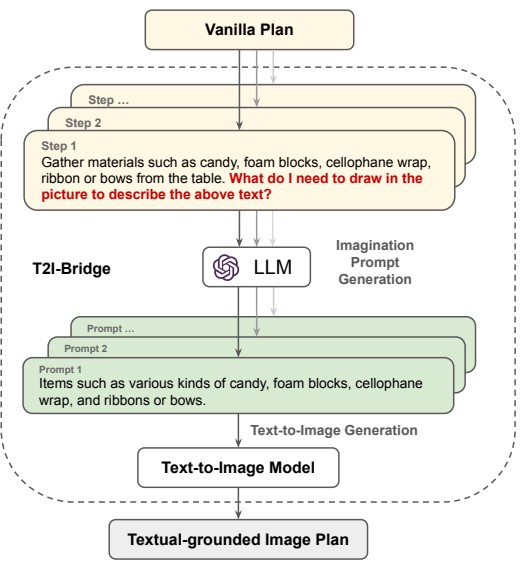

Figure 3: T2I-B for textual-grounded image plan.

**Text-to-Image Generation** We exploit the Stable Diffusion (Rombach et al., 2022) model to generate RGB images at $512 \times 512$ resolution. Figure 4 provides examples of text-to-image generation with and without our T2I-B. Benefiting from the existing knowledge in LLMs, the text-to-image models are able to generate semantically relevant and high-fidelity images based on the already processed prompt.

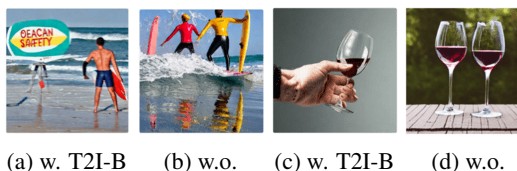

(a) w. T2I-B    (b) w.o.    (c) w. T2I-B    (d) w.o.

Figure 4: Text-to-image generation showcases with (a) (b) on "before hitting the waves, read up on ocean safety tips and know the rules of the beach" and (c) (d) on "put down the wine glass" with or without T2I-B.

### 3.5 VISUAL-GROUNDED TEXT PLAN GENERATION WITH IMAGE-TO-TEXT BRIDGE

To enhance the completeness, alignment, and knowledge exchange between the generated text and image plans, we propose revising the vanilla text plan using the textual-grounded image plan.

**Image Verbalization** To complete this, we first need to transfer the visual plan into a natural language format and then inject it into LLM. We implement this by generating captions for each visual plan. Given the image $v$, the captioning model BLIP (Li et al., 2022) generates captions, which transfer the visual knowledge into textual descriptions. For each generated visual plan $v_i$ at each timestep $i$, we generate such pairwise caption with $caption = G(v, Desc)$, where $Desc$ denotes the task description for unified vision and language models, "what does the image describe" in our case. With these image captions, we can further transfer the visual-grounded information into LLM and revise our textual plan.

**Revision Generation** To ground the textual plan in visual context, we use the verbalized description of the visual plan to concatenate with our Image-to-Text Bridge template similar to "Let's revise the procedure using the captions". Concretely, we concatenate the initial textual plan, the captions of the visual plan, and the Image-to-Text Bridge template as the prompt $P_{i2t}$ "Step-by-step Procedure: *[INITIAL]* Captions: *[CAPTION] [I2T-B]*". In this way, we elicit the reasoning ability of LLMs to ground the textual plan in verbalized visual context, as depicted in Figure 5 To this end, our generated multimodal plan is bi-directional grounded by connecting the abilities of LLMs and multimodal generative models.

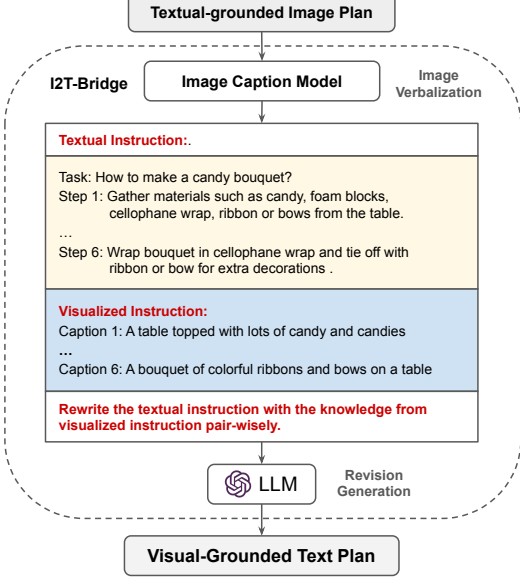

Figure 5: I2T-B injects verbalization of the image plans to foster revision generation of visual-grounded text plans with awareness of multimodal context.

## 4 EXPERIMENTS

### 4.1 DATASETS

Our datasets are collected and repurposed from WIKIHOW[2] and RECIPEQA(Yagcioglu et al., 2018) due to their temporal relatedness among texts and images. We collect WIKIPLAN by crawling the household "how to" articles from WIKIHOW and then repurpose them into a multimodal procedural planning dataset by formulating the article title as the task name and content as the textual steps, with

---

[2]https://www.wikihow.com

Table 1: Percentages of multimodal procedural planning results of TIP that are better than, tied with, or worse than baselines, on randomly sampled 200 distinct tasks from each dataset.

| Dataset | Ours vs. Model | Textual-Informativeness | | | Visual-Informativeness | | | Temporal Coherence | | | Plan Accuracy | | |
|---|---|---|---|---|---|---|---|---|---|---|---|---|---|
| | | Win(↑) | Tie | Lose(↓) | Win(↑) | Tie | Lose(↓) | Win(↑) | Tie | Lose(↓) | Win(↑) | Tie | Lose(↓) |
| WIKIPLAN | Image Ref + OFA-Caption | **63.34** | 18.38 | 18.27 | **60.63** | 20.45 | 18.92 | **61.95** | 21.03 | 17.02 | **61.99** | 19.40 | 18.61 |
| | Image Ref + BLIP-Caption | **62.70** | 18.70 | 18.60 | **61.26** | 21.18 | 17.56 | **62.22** | 20.78 | 17.00 | **62.29** | 18.28 | 19.43 |
| | Text Ref + DALLE | **62.61** | 20.34 | 17.06 | **59.88** | 22.38 | 17.74 | **60.53** | 22.08 | 17.40 | **61.19** | 22.07 | 16.74 |
| | Text Ref + Stable-Diffusion | **62.58** | 19.82 | 17.60 | **60.25** | 21.16 | 18.58 | **60.68** | 22.38 | 16.94 | **61.73** | 20.56 | 17.72 |
| | Text-Davinci-002 + Stable-Diffusion | **60.68** | 21.56 | 17.76 | **59.90** | 20.41 | 19.70 | **60.22** | 22.99 | 16.79 | **60.41** | 21.53 | 18.06 |
| | Text-Davinci-003 + Stable-Diffusion | **62.32** | 19.82 | 17.86 | **60.29** | 20.85 | 18.85 | **61.10** | 22.17 | 16.73 | **61.48** | 20.29 | 18.23 |
| RECIPEPLAN | Image Ref + OFA-Caption | **64.51** | 18.29 | 17.20 | **62.39** | 20.18 | 17.43 | **62.74** | 20.40 | 16.86 | **63.66** | 19.19 | 17.15 |
| | Image Ref + BLIP-Caption | **64.81** | 18.58 | 16.61 | **62.29** | 19.60 | 18.11 | **62.70** | 20.72 | 16.58 | **62.90** | 19.08 | 18.02 |
| | Text Ref + DALLE | **61.16** | 20.15 | 18.69 | **59.60** | 20.60 | 19.80 | **60.04** | 20.48 | 19.48 | **62.11** | 19.21 | 18.68 |
| | Text Ref + Stable-Diffusion | **61.31** | 19.81 | 18.87 | **60.49** | 20.37 | 19.14 | **60.37** | 20.33 | 19.31 | **62.38** | 18.81 | 18.81 |
| | Text-Davinci-002 + Stable-Diffusion | **62.50** | 19.33 | 18.17 | **60.59** | 18.12 | 21.29 | **61.24** | 21.13 | 17.63 | **62.30** | 17.38 | 20.31 |
| | Text-Davinci-003 + Stable-Diffusion | **62.65** | 19.26 | 18.09 | **61.10** | 20.00 | 18.90 | **61.46** | 20.60 | 17.94 | **62.85** | 18.75 | 18.40 |

the pictures as the visual steps. We collect RECIPEPLAN from RECIPEQA dataset for multimodal procedural planning by sequencing all the given text-image pairs as the text and image plan correspondingly, with the main title as the task name. We conduct zero-shot experiments on $1,000$ distinct, randomly sampled tasks from each dataset. More dataset details are in Appendix C.

Table 2: Automatic evaluations on $2,000$ distinct tasks from WIKIPLAN and RECIPEPLAN. Image Ref and Text Ref baselines use image and text title references from the dataset. Our TIP uses Text-Davinci-003 and Stable-Diffusion as the LLM and T2I model. We underline and **bold** highest score of models with and without reference baselines.

| Dataset | Model | Text Plan | | | | Image Plan | | Multimodality Plan | | | Step Length |
|---|---|---|---|---|---|---|---|---|---|---|---|
| | | WMD | S-BERT | ROUGE-L | METEOR | FID↓ | CLIP↑ | Cap-S | Text-S | ALL-S | Avg. |
| WIKIPLAN | Image Ref + BLIP-Caption | 0.78 | 0.35 | 0.06 | 0.04 | - | 0.71 | 0.36 | 0.41 | 0.39 | 8.26 |
| | Image Ref + OFA-Caption | 0.86 | 0.27 | 0.07 | 0.06 | - | 0.71 | 0.27 | 0.48 | 0.37 | 8.26 |
| | Text Ref + DALLE | 0.68 | 0.76 | 0.28 | 0.12 | 47.39 | 0.74 | 0.33 | 0.26 | 0.29 | 8.26 |
| | Text Ref + Stable-Diffusion | 0.68 | 0.76 | 0.28 | 0.12 | 56.64 | 0.73 | 0.34 | 0.26 | 0.30 | 8.26 |
| | Text-Davinci-002 + Stable-Diffusion | 0.87 | 0.65 | 0.10 | 0.06 | 61.17 | 0.50 | 0.33 | 0.25 | 0.28 | 4.70 |
| | Text-Davinci-003 + Stable-Diffusion | 0.86 | 0.67 | 0.11 | 0.08 | 57.87 | 0.70 | 0.33 | 0.27 | 0.30 | 6.68 |
| | **TIP (Ours)** | **0.90** | **0.67** | **0.12** | **0.09** | **48.82** | **0.78** | **0.34** | **0.28** | **0.31** | 6.75 |
| RECIPEPLAN | Image Ref + BLIP-Caption | 0.77 | 0.37 | 0.08 | 0.05 | - | 0.64 | 0.42 | 0.56 | 0.49 | 6.93 |
| | Image Ref + OFA-Caption | 0.82 | 0.40 | 0.09 | 0.10 | - | 0.64 | 0.43 | 0.48 | 0.46 | 6.93 |
| | Text Ref + DALLE | 0.21 | 0.59 | 0.10 | 0.09 | 53.55 | 0.63 | 0.46 | 0.40 | 0.43 | 6.93 |
| | Text Ref + Stable-Diffusion | 0.21 | 0.59 | 0.10 | 0.09 | 54.58 | 0.61 | 0.48 | 0.40 | 0.44 | 6.93 |
| | Text-Davinci-002 + Stable-Diffusion | 0.84 | 0.63 | 0.11 | 0.10 | 60.11 | 0.49 | 0.44 | 0.33 | 0.38 | 5.17 |
| | Text-Davinci-003 + Stable-Diffusion | 0.85 | 0.68 | 0.12 | 0.13 | 60.07 | 0.73 | 0.42 | 0.35 | 0.38 | 6.82 |
| | **TIP (Ours)** | **0.86** | **0.68** | **0.13** | **0.14** | 58.68 | **0.79** | **0.43** | **0.36** | **0.40** | 6.94 |

## 4.2 EVALUATION METRICS

We conduct head-to-head comparisons using Amazon Mechanical Turk (AMT) platform (details can be found in Appendix D.1) on four aspects: (1) `Textual Informativenss`: the text plans contain the necessary information to complete the task, (2) `Visual Informativeness`: the image plans contain the necessary information to complete the task, (3) `Temporal Coherence`: the multimodal plans meet the temporal commonsense requirements, such as the order in which the steps occur, (4) `Planning Accuracy`: whether referring to the multimodal plans can successfully assist task completion. In addition, we measure semantic relevance between predicted text plans and reference text plans using Word Mover's Distance (WMD) (Kusner et al., 2015), Sentence-BERT (S-BERT) (Reimers & Gurevych, 2019), ROUGE-L (Lin, 2004), and METEOR (Banerjee & Lavie, 2005). We evaluate image plans using FID (Heusel et al., 2017) and CLIPScore (Hessel et al., 2021; Radford et al., 2021). We calculate Caption-Sentence-BERT (Cap-S) and Text-Sentence-BERT (Text-S) scores by comparing predicted image and text plans respectively to reference text using S-BERT. The average of these yields the All-Sentence-BERT (ALL-S) score for multimodal plans. Evaluations are conducted at a procedure level. More details of evaluation can be found in Appendix D.

## 4.3 BASELINES

As we are exploring a novel task, that require generating the image and text plans conditioned on the high-level goals, no existing models can be directly applied. To validate the effectiveness of our designed dual bridges, we devise several intuitive baselines for comparison: (1) ImageRef +

Table 3: Robustness check of various templates used in both Text-to-Image Bridge and Image-to-Text Bridge over WIKIPLAN and RECIPEPLAN dataset. The underlined templates are misleading examples. Our Text-Image Prompting model chooses the template with averaged best multimodal alignment, highlighted in purple.

| Text-to-Image Bridge Template | Alignment | | Image-to-Text Bridge Template | Alignment | |
|---|---|---|---|---|---|
| | WIKIPLAN | RECIPEPLAN | | WIKIPLAN | RECIPEPLAN |
| What do I need to draw in the picture to describe the above text? | **0.9625** | **0.9595** | Rewrite the textual instruction with the knowledge from visualized instruction pair-wisely. | 0.7644 | 0.6945 |
| What do you see in the figure? | 0.9366 | 0.9397 | Based on the visual caption, can you revise the step-by-step procedure according to the paired captions? | **0.8011** | 0.6205 |
| Describe what the picture corresponding to the text should have. | 0.9070 | 0.9181 | Revise each step according to the visual imagination. | 0.6921 | 0.7329 |
| Let's think about what we need to visualize to present the above idea. | 0.8986 | 0.8941 | Let's revise the procedure using the captions. | 0.6155 | **0.7691** |
| Describe something irrelevant to the above text. | 0.5598 | 0.5325 | What's the procedure that disobey the captions? | 0.5079 | 0.5902 |
| What do you usually draw? | 0.5350 | 0.4826 | Provide an interesting procedure to be irrelevant with the captions. | 0.1519 | 0.163 |

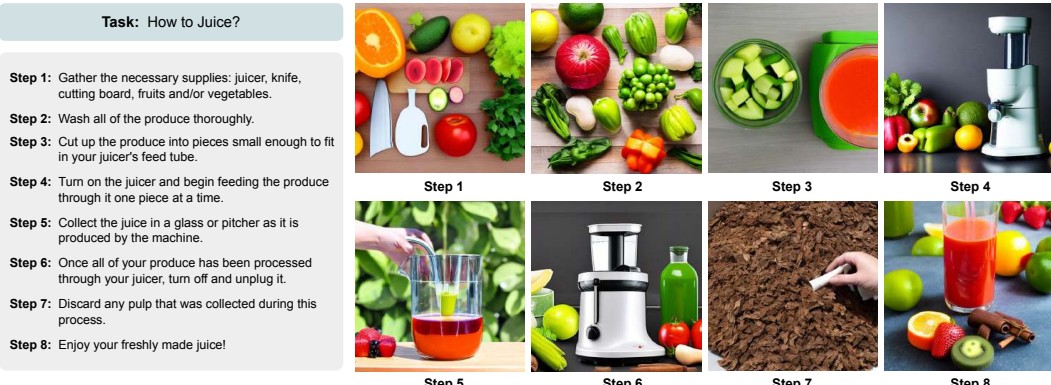

(a) Multimodal procedural plan generated by baseline Text-Davinci-003 + Stable-Diffusion.

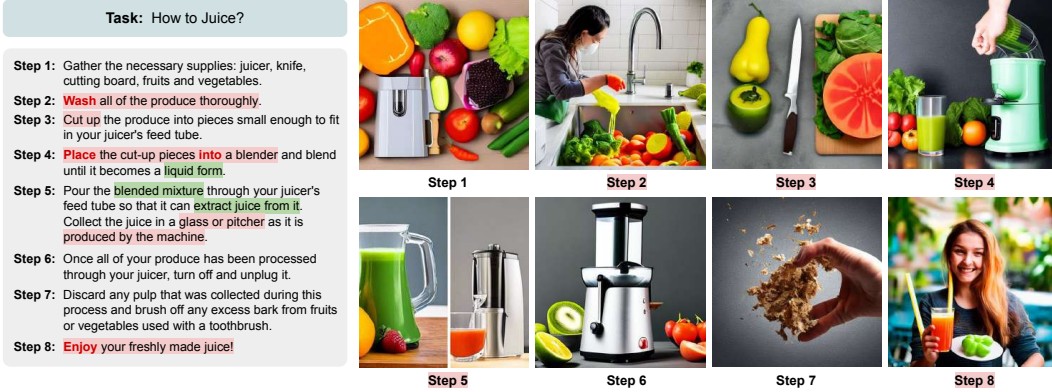

(b) Multimodal procedural plan generated by our Text-Image Prompting (TIP).

Figure 6: Improved grounding in textual and visual context are highlighted in pink and green respectively. Red texts indicate reasoning of physical action in image plan generation.

OFA/BLIP-Caption: use image sequences from each article/recipe as the image plans references (ImageRef), and extract captions of image references as text plans using image caption models (e.g., OFA or BLIP) (2) TextRef + DALLE/Stable-Diffusion: use titles from each article/recipe as text plans references (TextRef), and use text-to-image models (e.g., DALLE or Stable-Diffusion) to generate images as image plans conditioned on text references (3) Text-Davinci-002/003 + Stable-Diffusion: given the high-level goal, first prompt the LLMs (e.g., Text-Davinci-002/003) to generate text plans, and then use text-to-image models to generate image plans conditioned on the text plans (4) Text-Davinci-003 (Step-based) + Stable-Diffusion: all the variants are in default procedure-based, however this variant instead of generating the plan at the procedure level, it generates one step of the text plans at a time iteratively. These variants are using LLMs and T2I models separately without collaboration.

### 4.4 QUANTITATIVE ANALYSIS

**Human Evaluation Results** We conduct Win-Tie-Lose Comparison between TIP and the baselines over WIKIPLAN and RECIPEPLAN. Averaged results from 200 tasks rated by 3 crowdsourcing per example are reported in Table 1. Across four aspects, TIP receives consistently higher preferences, outperforming the baselines over the winning ratio by over 60%. In terms of textual informativeness, the unimodal baselines (Image Ref + OFA-Caption and Image Ref + BLIP-Caption) is slightly worse than the unimodal text reference-based baseline (Text Ref + Stable-Diffusion and Text Ref + DALLE) and multimodal baselines (Text-Davinci-003 + Stable-Diffusion and Text-Davinci-002 + Stable-Diffusion). This is mainly due to the other baselines either directly leveraging the textual information from the reference or the rich text-based knowledge in LLMs. In terms of visual informativeness, the multimodal baselines (Text-Davinci-003 + Stable-Diffusion and Text-Davinci-002 + Stable-Diffusion) can not achieve on-par results with textual reference-based baseline. We hypothesize this is due to the lack of visual knowledge injected into LLMs. The performance gain of TIP over multimodal baselines (Text-Davinci-003 + Stable-Diffusion and Text-Davinci-002 + Stable-Diffusion) imply the importance of grounding our multimodal plans in a multimodal context.

**Automatic Evaluation Results** In Table 2, TIP achieves consistent improvement over baselines (without Ref), and even surpasses the baselines using reference from the dataset on RECIPEPLAN. This further confirms our superiority in generating multimodal plans with semantic correctness and alignment. Notice that Text Ref baselines directly use the title from the dataset, which is a summarized version of the main content (golden reference used in automatic evaluations).

**Template Robustness** In Table 3, we compare various similar templates for T2I-B and I2T-B against misleading templates. The Alignment is measured with CLIP (Radford et al., 2021) to capture the similarity between given text/image and conditionally generated image/text. The poor alignment of misleading templates and similar alignment of various bridge templates prove the robustness of the template choice in the experiments.

**Single Bridge Effect** We report the effects of Text-to-Image Bridge and Image-to-Text Bridge of TIP in Table 4. The performance drop indicates that the text plan without condition on visual information is vulnerable in text-only planning quality. In Table 5, we also observe using T2I-B brings obvious improvement over both FID score and Alignment of image plans. These results further indicate that the key ingredient of our proposed method TIP is that LLMs and multimodal generative models will collaboratively generate multimodal procedural plans benefiting from our designed dual bridges.

Table 4: Effects of Text-to-Image Bridge and Image-to-Text Bridge over text plan generation.

| Model | WIKIPLAN | RECIPEPLAN |
|---|---|---|
| | Avg. Textual | Avg. Textual |
| w.o. T2I-B | 0.341 (-18.4%) | 0.363 (-14.1%) |
| w.o. I2T-B | 0.261 (-37.5%) | 0.273 (-35.4%) |

Table 5: Effects of Text-to-Image Bridge on image plan generation.

| Model | WIKIPLAN | | RECIPEPLAN | |
|---|---|---|---|---|
| | FID ↓ | Align ↑ | FID ↓ | Align ↑ |
| Text Ref + DALLE | 119.03 | 0.77 | 83.27 | 0.64 |
| + T2I-B | **117.02** | **0.79** | **67.64** | **0.78** |
| Text Ref + Stable Diffusion | 129.13 | 0.74 | 88.17 | 0.62 |
| + T2I-B | **119.74** | **0.78** | **84.37** | **0.78** |

**Step-based or Procedure-based** We explore our procedure-based method (P-Ours) against the step-based TIP (S-Ours) and step-based Text-Davinci-003 + Stable-Diffusion (S-Base). The main difference from step-based approach is the plan is generated one step at a time. Since LLMs achieve promising long text reasoning capability, we generate the full procedure for efficiency consideration. We report the head-to-head comparison results in Figure 7. The procedure-based method achieves 60% win rate over the step-based TIP. We observe this is partially due to the instinct of LLMs to repeat input texts and is less clear to understand the full intent of generation expectation. Thus the procedure-based method usually achieves better planning quality at the very beginning.

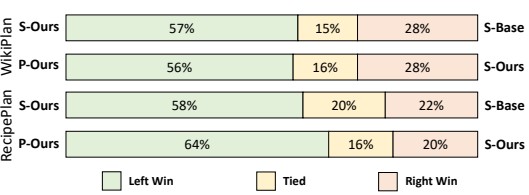

Figure 7: Step-based (S) vs. Procedure-based (P) Win-Tie-Lose over `Plan Accuracy`.

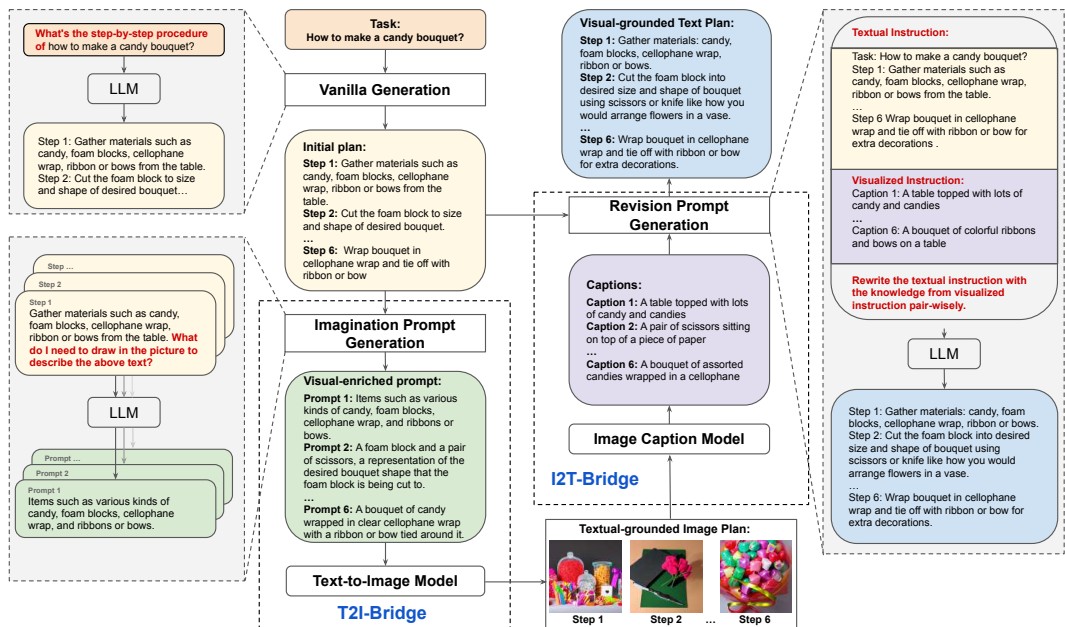

Figure 8: Output Details of Each Module of TIP for Multimodal Procedural Planning. The showcase task is "How to make a candy bouquet" sampled from RECIPEPLAN.

## 4.5 QUALITATIVE ANALYSIS

**Multimodal Grounding** In Figure 6, we compare the performance of TIP to baselines in multimodal procedural planning. TIP generate image plans that are grounded in the textual context. With the help of LLMs reasoning in the temporal dimension, we transfer this ability to image generation, conditioning on the revised prompts of LLMs. This allows digestion of the temporal and complex reasoning present in the text plan and directly indicates what needs to be depicted in the image. The highlighted steps of image plans correctly visualize the scene described in the textual context. For example, at Step 2, instead of only showing the vegetables, ours show an image of a person washing the produce thoroughly. TIP also generate text plans that are better grounded in the image plan. The text plan correctly refers to the objects in visual input, such as "liquid form" and "blended mixture", and also complements the visual context, such as "extract juice from it". Our results indicate the potential for uncovering multimodal reasoning capabilities in LLMs. We provide more comparisons on multimodal procedural planning in Appendix E.1.

**Module Outputs** We showcase all the details of the outputs of each module for the example task "How to make a candy bouquet" in Figure 8. The T2I-B leverage the complex language comprehension and zero-shot reasoning ability of LLMs to generate imagination prompt. This enhances the specificity of the initial plan in terms of improving text-to-image generation. With the textual-grounded image plan, we can extract higher-quality visual information to reversely revise the text plan. More specifically, the I2T-B injects visual knowledge via verbalization of the visual plans to generate a visually-grounded and complementary textual plan. In the end, the generated image and text plans are mutually grounded. Please refer to Appendix B.2 for more module output details.

## 5 CONCLUSION AND FUTURE WORK

We introduce the Multimodal Procedural Planning task that aims to generate goal-conditioned text and image subsequences and benchmark models' performance with our curated testbed WIKIPLAN and RECIPEPLAN. We propose Text-Image Prompt (TIP), a dual-modality prompting framework, that connects large language models with multimodal generative models to enable plausible multimodal procedural plan generation. We hope our work shed light on future research into uncovering this limitless capability of multimodal procedural planning driven by uniform automatic metrics.

## ETHICS STATEMENT

We acknowledge that our research utilizes resourceful knowledge in large-scale pre-trained models, which have the potential to bias to a certain cultural background. For example, the task from RECIPEPLAN and WIKIPLAN that involve food preparation may have different procedures depending on different individuals' eating habits. We encourage future studies that uncover the multimodal procedural planning ability with consideration of personalized decision makings.

The data annotation part of the project is classified as exempt by Human Subject Committee via IRB protocols. The hourly wage paid to participants is estimated at $12, which is higher than the federal minimum wage. We manually ensure no personal information is collected and no offensive content is presented during human evaluations.

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

# A  BACKGROUND

A line of work in unimodal procedural planning studies sorting a series of unordered texts or events Chen et al. (2016); Cui et al. (2018); Oh et al. (2019); Calizzano et al. (2021); Wu et al. (2022). Other work explores generating subsequent steps given a target goal, e.g., generating a sequence of plans to complete the high-level task (Lu et al., 2022a).

Text-to-image generation is a task that synthesizes images from text prompts. DALL-E 2 Ramesh et al. (2022) and Stable Diffusion Rombach et al. (2022) are state-of-the-art text-to-image models developed on top of diffusion models conditioned on input texts. Some early work in text-to-image models trains generative adversarial networks (GANs) Goodfellow et al. (2014) on image captioning datasets Xu et al. (2018); Zhu et al. (2019); Tao et al. (2020); Zhang et al. (2021a); Ye et al. (2021) to generate images conditioned on textual descriptions. Other work follows the VQ-VAE Van Den Oord et al. (2017) framework and trains autoregressive transformers that take both the text and image as sequences of tokens Ramesh et al. (2021); Ding et al. (2021); Gafni et al. (2022). However, these methods are struggling to generate photorealistic images. Motivated by the remarkable progress of diffusion models in generating images with fidelity Sohl-Dickstein et al. (2015); Song & Ermon (2019); Ho et al. (2020), recent work has applied them to text-to-image generation with auxiliary text encoders Rombach et al. (2022); Nichol et al. (2021); Gu et al. (2022); Ramesh et al. (2022); Saharia et al. (2022). In Wang et al. (2022e), they propose the first large-scale text-to-image prompt dataset, DiffusionDB, which enables a new research direction of prompt engineering to construct better prompts. In Chakrabarty et al. (2022), they use GPT-3 to generate a detailed textual description with rich visual metaphors to prompt the DALL-E 2 model.

Recently, there has been a trend of using large language models (LLMs) like GPT-3 Brown et al. (2020) to transfer visual knowledge in order to improve their capabilities in downstream natural language processing (NLP) and multimodal tasks. For example, images and videos can be translated into captions which further instruct a language model to generate contextual descriptions Wang et al. (2022d); Zeng et al. (2022) or answer knowledge-based visual questions Yang et al. (2022c). Instead of being prompted with textual descriptions, language models can extend to vision-language settings through text generation controlled by visual features Cho et al. (2021); Tsimpoukelli et al. (2021); Su et al. (2022); Zhu et al. (2022b); Wang et al. (2022b); Alayrac et al. (2022).

# B  METHOD DETAILS

## B.1  CONFIGURATIONS

The experiments using Text-Davinci and DALLE are conducted with OpenAI API on January 2023. We use BLIP w/ ViT-B and CapFilt-L[3] and OFA-base from huggingface demo[4].

## B.2  DETAILS OF MODULE OUTPUTS

We visualize the output details of each module in Figure 9 and Figure 10.

# C  DATASET DETAILS

## C.1  RECIPEPLAN

**Data Repurpose** RECIPEQA was proposed in (Yagcioglu et al., 2018) that provide four tasks (Textual Cloze, Visual Cloze, Visual Ordering, Visual Coherence) for multimodal machine comprehension of cooking recipes. This dataset contains question-answer pairs generated from copyright-free recipes. Each of them is under a license, which is provided in each data JSON file. We collect RECIPEPLAN by repurposing the test dataset from RECIPEQA that relates to the Visual Ordering task as the sequence generation task for the multimodal procedural planning evaluation testbed. We use recipe instructions as textual plan reference and their paired images as visual plan reference.

---

[3]https://github.com/salesforce/BLIP
[4]https://huggingface.co/OFA-Sys/ofa-base

**Task:** How to make a candy bouquet?

| Vanilla Text plan | Vanilla Image Plan | Imagined Prompt | Textual-Grounded Image Plan | Verbalization | Visual Grounded Text plan |
|---|---|---|---|---|---|
| **Step 1:** Gather materials: candy, foam blocks, cellophane wrap, ribbon or bows. |  | Items such as various kinds of candy, foam blocks, cellophane wrap, and ribbons or bows. |  | A table topped with lots of candy and candies | **Step 1:** Gather materials such as candy, foam blocks, cellophane wrap, ribbon or bows from the table. |
| **Step 2:** Cut the foam block to size and shape of desired bouquet. |  | A foam block and a pair of scissors, a representation of the desired bouquet shape that the foam block is being cut to. |  | A pair of scissors sitting on top of a piece of paper | **Step 2:** Cut the foam block into desired size and shape of bouquet using scissors or knife like how you would arrange flowers in a vase. |
| **Step 3:** Insert wooden skewers into each piece of candy. |  | Wooden skewers inserted into each piece of candy, arrows pointing to the skewers to indicate that they are being inserted into the candy. |  | A group of toothbrushes sitting on top of a table | **Step 3:** Insert wooden skewers into each piece of candy as if they were lollipops on top of a blue table. |
| **Step 4:** Arrange the pieces of candy onto the foam in a pleasing pattern. |  | A bouquet of candy in a pleasing pattern. The candy should be arranged on top of a foam base. The candy should be arranged in an aesthetically pleasing pattern, such as alternating colors, sweet and |  | A bouquet of colorful candies in a vase | **Step 4:** Arrange the pieces of candy onto the foam in an interesting pattern like playing with colored rocks on plate and cupcakes and candies for your bouquet decoration ideas. |
| **Step 5:** Secure pieces with hot glue if necessary. |  | A bouquet of candy pieces that are being held together with hot glue, a person holding a hot glue gun, to show that the pieces of candy are being secured with hot glue. |  | A pile of different colored candies and lollipops | **Step 5:** Secure pieces with hot glue if necessary just like arranging colorful ribbons and bows around the bouquets centerpiece. |
| **Step 6:** Wrap bouquet in cellophane wrap and tie off with ribbon or bow. |  | A bouquet of candy wrapped in clear cellophane wrap with a ribbon or bow tied around it. |  | A bouquet of assorted candies wrapped in a cellophane | **Step 6:** Wrap bouquet in cellophane wrap and tie off with ribbon or bow for extra decorations. |

(a) Full example outputs.

Figure 9: Output Table Details of each module of TIP for Multimodal Procedural Planning.

**Dataset Statistics** We visualize two examples of our repurposed RECIPEPLAN for multimodal procedural planning in Figure 11. We also show the word-cloud distribution of task name and textual plan reference in Figure 13.

## C.2 WIKIPLAN

**Raw Data Collection** To facilitate research on learning to generate procedural planning in a multimodal setting, we have constructed the large-scale WIKIPLAN dataset collected from the WIKIHOW website[5], which is under an Attribution-Noncommercial-Share Alike 3.0 Creative Commons License.. This website provides a wide range of how-to articles related to everyday life topics, which are collaboratively written by its users and reviewed by experts. We crawled each article, collecting the task title, URL, introduction, topics, and steps. Each step includes a brief textual action, a detailed context, and an illustration image. Our raw dataset consists of 30,026 examples across 19 categories and 2,062 topics. We plan to release the raw data in the hopes of pre-training models for procedural planning and knowledge reasoning.

**Quality Control** To improve the evaluation of different baselines, we further selected five categories that feature temporal actions and high-fidelity visual contexts: *Food and Entertaining*, *Hobbies and Crafts*, *Home and Garden*, *Pets and Animals*, and *Sports and Fitness*. In order to reflect common tasks in real-life scenarios, articles with fewer than three steps or more than 22 steps were excluded, as well as articles with images of a dimension size of fewer than 400 pixels. Each category was balanced with 200 examples. To further ensure high quality, we conducted a quality control in which well-trained human annotators reviewed the dataset and manually revised the examples if there was any wrong or inappropriate content.

**Dataset Statistics** Finally, our WIKIPLAN dataset consists of 1,000 examples across 5 categories and 370 different topics. Three examples of the dataset are illustrated in Figure 12, which include muddling mint leaves for a cocktail, encouraging a cat to eat, and becoming a better football player.

---

[5]https://www.wikihow.com/

**Task:** How to make peppermint patty pudding shot?

| Vanilla Text plan | Vanilla Image Plan | Imagined Prompt | Textual-Grounded Image Plan | Verbalization | Visual Grounded Text plan |
|---|---|---|---|---|---|
| **Step 1:** Combine 1/2 cup of peppermint schnapps, 1/4 cup of Bailey's Irish cream, and 2 tablespoons of chocolate syrup in a bowl or blender. | | A bowl or blender filled with 1/2 cup of peppermint schnapps, 1/4 cup of Bailey's Irish cream, and 2 tablespoons of chocolate syrup. | | Two cups of hot chocolate with spoons on a table | **Step 1:** Place three cups of chocolate pudding on a flat surface and add 1/2 cup of peppermint schnapps, 1/4 cup of Baileys Irish Cream, and 2 tablespoons of chocolate syrup. |
| **Step 2:** Add 4 ounces of cold heavy whipping cream to mixture and blend until smooth. | | A bowl with the mixture in it, a spoon in the mixture, a measuring cup with 4 ounces of heavy cream, a blender, and a glass with the peppermint patty pudding shot. | | A table topped with three different types of desserts. | **Step 2:** Using a spoon or blender mix all ingredients until smooth. |
| **Step 3:** Place 4 small glasses on a flat surface and fill with pudding shot mix. | | Four small glasses on a flat surface, each filled with a pudding shot mix. You should also include any relevant ingredients, such as peppermint extract, liqueur, and cream, to complete the pudding. | | A table topped with three shot glasses filled with liquid. | **Step 3:** Pour the mixture into four small glasses and top each with whipped cream. |
| **Step 4:** Chill for at least 30 minutes before serving. | | A picture of a tray of peppermint patty pudding shots in the refrigerator, with a timer counting down the 30 minutes. | | A tray filled with chocolate cups and candy canes. | **Step 4:** Add a candy cane for garnish on each glass and chill for at least 30 minutes before serving. |

**Task:** How to make bartending for beginners simple margarita?

| Vanilla Text plan | Vanilla Image Plan | Imagined Prompt | Textual-Grounded Image Plan | Verbalization | Visual Grounded Text plan |
|---|---|---|---|---|---|
| **Step 1:** Gather the following ingredients: tequila, triple sec (or Cointreau), lime juice, and simple syrup. | | A glass of margarita with a lime wedge garnish, and next to it illustrate the ingredients needed to make the margarita - a bottle of tequila, bottle of triple sec or Coint | | A bottle of orange juice next to limes and lime slices. | **Step 1:** Gather the following ingredients: tequila, triple sec (or Cointreau), lime juice, and simple syrup. |
| **Step 2:** Fill a shaker with ice cubes. | | A shaker filled with ice cubes. | | A glass filled with ice and lime slices. | **Step 2:** Fill a shaker with ice cubes. |
| **Step 3:** Pour in 1 ounce of tequila, ½ ounce of triple sec or Cointreau, ¾ ounces of fresh lime juice, and ½ ounce of simple syrup into the shaker filled with ice cubes. | | A shaker filled with ice cubes, a bottle of tequila, a bottle of triple sec or Cointreau, a bottle of fresh lime juice, and a bottle of simple syrup. | | A glass of lemonade next to a bottle of lemonade. | **Step 3:** Pour in 1 ounce of tequila, ½ ounce of triple sec or Cointreau, ¾ ounces of fresh lime juice into the shaker filled with ice cubes and shake well until all ingredients are combined and chilled (about 20 seconds). |
| **Step 4:** Shake well until all ingredients are combined and chilled (about 20 seconds). | | A person shaking a cocktail shaker filled with ice cubes, limes, tequila, and triple sec. You could also add a timer set to 20 seconds to illustrate how long the drink should be shaken. | | A person pouring a drink into a glass. | **Step 4:** Add half an ounce of simple syrup before straining the mixture into a glass rimmed with salt or sugar-lime mix (optional). |
| **Step 5:** Strain the mixture into a glass rimmed with salt or sugar-lime mix (optional). | | A glass with a rim lined with either salt or a sugar-lime mix and a pitcher or shaker filled with a mixture of margarita. A spoon or strainer. | | A glass of water with a lime next to it. | **Step 5:** Garnish with a wedge of lime or orange slice before serving cold margarita in your desired glass! |
| **Step 6:** Garnish with a wedge of lime or orange slice before serving cold margarita to your guests! | | A glass filled with a cold margarita, a wedge of lime or orange slice on the side of the glass, and possibly some straws. | | A glass of orange juice with a straw and lime. | **Step 6:** Enjoy your delicious homemade drink! |

Figure 10: Full example of outputs details of TIP for Multimodal Procedural Planning.

Each example is composed of a title, introduction, related topics, and a list of detailed steps with visual aids. The word-cloud distributions of the task titles and step text are visualized in Figure 14, which demonstrate the diversity of topics and words.

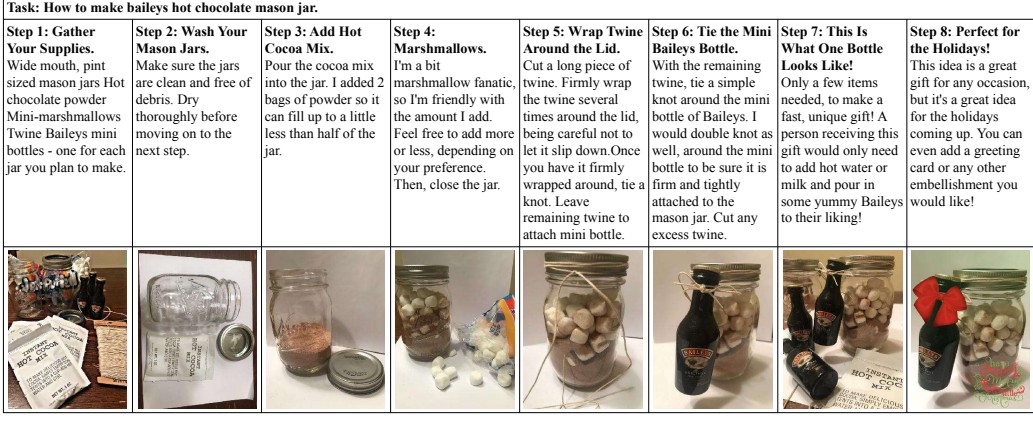

Figure 11: Two examples in the RECIPEPLAN dataset.

## D    EVALUATION DETAILS

### D.1    CROWDSOURCING HUMAN EVALUATION

We manually ensure no personal information is collected and no offensive content is presented during human evaluations. The hourly wage paid to participants is estimated at $12. And the total amount spent on participant compensation is $1958.

We average the results from 3 annotators for each example. Given the high-level goal (task name) for each assignment, we want the annotators to compare two generated text and image sequences in terms of *Textual-Informativeness*, *Visual-Informativeness*, *Temporal Coherence* and *Plan Accuracy*. Before going to the question, we let the annotators read the instructions below:

**Instruction:** Given the Task (e.g, Task: How to muddle), please compare two sequences of steps Sequence 1 and Sequence 2, and determine which sequence is better in terms of four aspects:

- Textual-Informativeness: whether the textual sequence (the sequence of texts) contains the amount of information needed to complete the task.
- Visual-Informativeness: whether the visual sequence (the sequence of images) contains the amount of information needed to complete the task.
- Temporal Coherence: whether the multimodal sequence (the paired sequence of texts and images) meets the temporal commonsense requirements, such as a step occurring before another step instead of after.
- Plan Accuracy: whether the multimodal sequence (the paired sequence of texts and images) can successfully complete the task.

To be concrete, the annotators were asked to choose one from the two sequences by *1 - Sequence 1 is better*, *2 - Tie*, and *3 - Sequence 2 is better*. We provide the multimodal plans as follows:

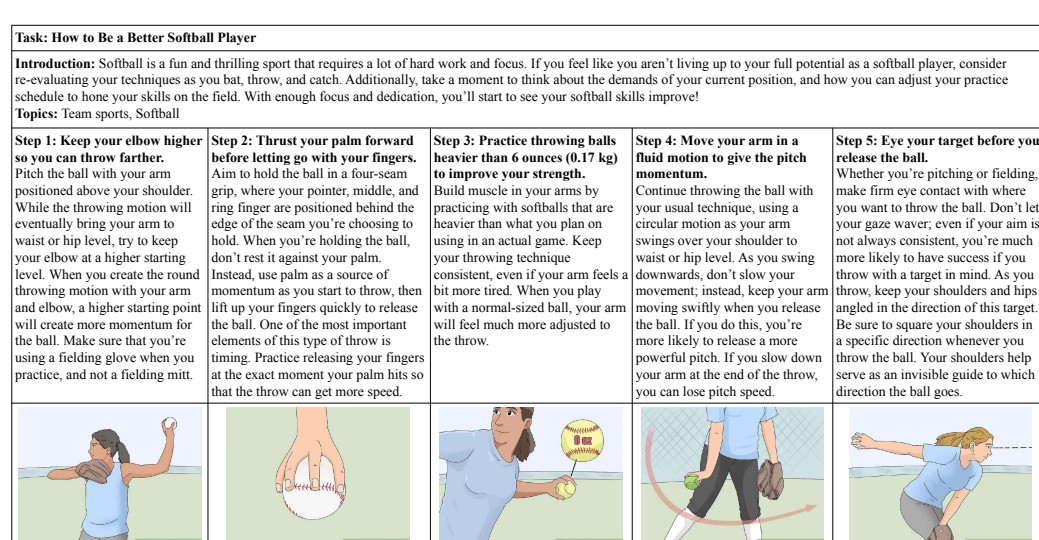

**Task: How to Muddle**

**Introduction:** Muddling is a bartender's technique for releasing flavor from solid cocktail ingredients. The basic idea is as simple as crushing the fruit or herb, but there are devils in the details ready to make your cocktail bitter or unpleasant. Learn how to do it the right way, and you won't have to muddle through on your own intuition.
**Topics:** Spirits and liqueurs, Cocktails

| **Step 1: Choose a gentle muddling tool.** A small, flat wooden tool is ideal, such as the end of a wooden spoon or a French rolling pin (without handles). Plastic or hard rubber tools will also work, but require a delicate touch. Avoid muddlers with teeth, since these tear up the leaves too much. A tough herb such as rosemary needs more breaking down. Follow the instructions for fruit instead. For instance, try using the wide, flat end of a wooden spoon. That will crush whatever you're muddling, without destroying it. | **Step 2: Put the ingredients in a sturdy glass.** Do not muddle in a delicate glass that may chip or break when struck. If the cocktail also calls for fruit, cucumber, or other non-herb ingredients, muddle them separately for best results. Granulated sugar tears into the ingredients as you muddle. This can be overkill for soft herbs, so add it with the fruit instead, or dissolve it in a few drops of water and add it to the cocktail separately. | **Step 3: Press and twist lightly.** Mint, basil, and other soft leaves release bitter flavors when crushed or torn apart. Just press down gently with the blunt tool while you twist your wrist, then release. Do this two or three times. Use your dominant hand to muddle while you hold the glass steady with your other hand. You only need to lightly bruise the leaves to release the oils and aroma. | **Step 4: Finish the drink.** The leaves are ready when lightly bruised, but still intact. You should be able to smell the herb, since the goal of muddling is to release flavorful, aromatic oils. You can leave the herbs in the final cocktail for presentation or strain them out, as desired. |
|---|---|---|---|

**Task: How to Encourage Your Cat to Eat**

**Introduction:** Cats are finicky creatures, and they may go off food when they are sick or getting older, for instance. If your cat suddenly stops eating, you should definitely take it to the vet for a checkup, as it may have an underlying problem. Otherwise, you can work on the cat, environment, and food to encourage your cat to eat, as well as use a few tricks when switching foods.
**Topics:** Cats, Feeding cats

| **Step 1: Feed the cat alone.** If there are other animals in the house, the cat may feel uncomfortable eating, as it tries to compete for food. If you section off a room where you can feed just that cat, it can increase the chances of the cat eating. | **Step 2: Give smaller meals.** Smaller meals throughout the day may seem counter-intuitive, but it can actually encourage your cat to eat. If your cat doesn't have a large appetite, it may be overwhelmed by a large bowl of food. | **Step 3: Offer comfort and attention.** Some cats want attention when they're eating. Try stroking your cat and talking soothingly to it while it's eating. You can also try a bit of petting if your cat is near the bowl but not eating. This tactic won't work with every cat, though, so if your cat seems disturbed by the attention, leave it alone. | **Step 4: Wipe the cat's nose.** If the cat has been sick, it may not be smelling very well. Wiping the cat's nose and trying to remove discharge may help it smell better. In turn, it may be more interested in food because it can smell it. | **Step 5: Hand feed the cat.** A cat who hasn't been interested in food may be more inclined to eat if you hand feed it. For canned food, you can put a little on your finger and offer it to the cat. For dry food, place a bit in your hand, and hold it out for the cat to eat. |
|---|---|---|---|---|

**Task: How to Be a Better Softball Player**

**Introduction:** Softball is a fun and thrilling sport that requires a lot of hard work and focus. If you feel like you aren't living up to your full potential as a softball player, consider re-evaluating your techniques as you bat, throw, and catch. Additionally, take a moment to think about the demands of your current position, and how you can adjust your practice schedule to hone your skills on the field. With enough focus and dedication, you'll start to see your softball skills improve!
**Topics:** Team sports, Softball

| **Step 1: Keep your elbow higher so you can throw farther.** Pitch the ball with your arm positioned above your shoulder. While the throwing motion will eventually bring your arm to waist or hip level, try to keep your elbow at a higher starting level. When you create the round throwing motion with your arm and elbow, a higher starting point will create more momentum for the ball. Make sure that you're using a fielding glove when you practice, and not a fielding mitt. | **Step 2: Thrust your palm forward before letting go with your fingers.** Aim to hold the ball in a four-seam grip, where your pointer, middle, and ring finger are positioned behind the edge of the seam you're choosing to hold. When you're holding the ball, don't rest it against your palm. Instead, use palm as a source of momentum as you start to throw, then lift up your fingers quickly to release the ball. One of the most important elements of this type of throw is timing. Practice releasing your fingers at the exact moment your palm hits so that the throw can get more speed. | **Step 3: Practice throwing balls heavier than 6 ounces (0.17 kg) to improve your strength.** Build muscle in your arms by practicing with softballs that are heavier than what you plan on using in an actual game. Keep your throwing technique consistent, even if your arm feels a bit more tired. When you play with a normal-sized ball, your arm will feel much more adjusted to the throw. | **Step 4: Move your arm in a fluid motion to give the pitch momentum.** Continue throwing the ball with your usual technique, using a circular motion as your arm swings over your shoulder to waist or hip level. As you swing downwards, don't slow your movement; instead, keep your arm moving swiftly when you release the ball. If you do this, you're more likely to release a more powerful pitch. If you slow down your arm at the end of the throw, you can lose pitch speed. | **Step 5: Eye your target before you release the ball.** Whether you're pitching or fielding, make firm eye contact with where you want to throw the ball. Don't let your gaze waver; even if your aim is not always consistent, you're much more likely to have success if you throw with a target in mind. As you throw, keep your shoulders and hips angled in the direction of this target. Be sure to square your shoulders in a specific direction whenever you throw the ball. Your shoulders help serve as an invisible guide to which direction the ball goes. |
|---|---|---|---|---|

Figure 12: Three examples in our curated WIKIPLAN dataset.

## Task: How to Get Kids to Eat Healthy.

### Sequence 1:

Step 1: Talk to your kids about the importance of eating healthy and make sure that nutritious food is accessible and visible in the house.

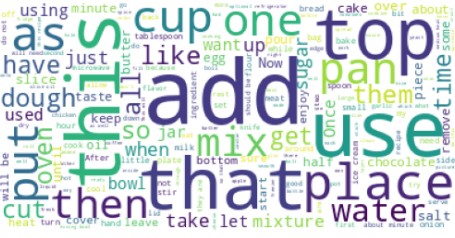

(a) RecipeQA step text

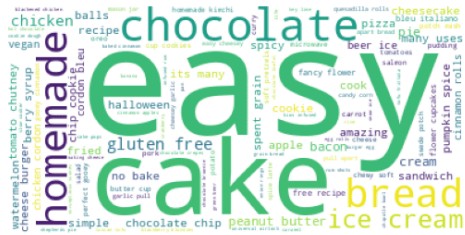

(b) RecipeQA task title text

Figure 13: Word cloud distributions of the task title and step text in the RECIPEPLAN dataset.

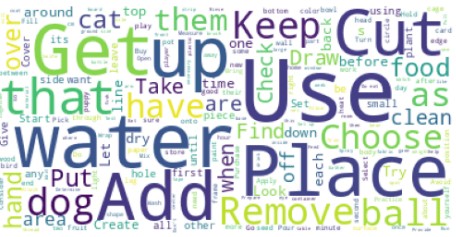

(a) Wikihow step text

(b) Wikihow task title text

Figure 14: Word cloud distributions of the task title and step text in the WIKIPLAN dataset.

Visual Plan at Step 1: [Image]

Step 2: Set rules or guidelines for what is allowed and not allowed in terms of snacks and meals., involve children in grocery shopping and meal preparation as much as you can, lead by example by practicing healthy eating habits yourself.

Visual Plan at Step 2: [Image]

Step 8: Encourage better dietary decisions at mealtimes by sitting down together with them at the table filled with various types of wholesome foods.

Visual Plan at Step 8: [Image]

**Sequence 2:**

Step 1: Talk to your kids about the importance of eating healthy.

Visual Plan at Step 1: [Image]

Step 2: Set rules or guidelines for what is allowed and not allowed in terms of snacks and meals.

Visual Plan at Step 2: [Image]

Step 8: Reward your child for trying new foods or making healthier choices.

Visual Plan at Step 8: [Image]

We show the paired visual plan at each step below the text plan.

Please refer to our Amazon Mechanical Turk human evaluation interface for head-to-head comparison on WIKIPLAN and RECIPEPLAN in Figure 16 and Figure 15 respectively.

## E    MORE RESULTS

### E.1    SHOWCASES

We show more cases in Figure 17- 20 comparing our Text-Image Prompting with powerful baselines Text-Davinci-003 + Stable-Diffusion.

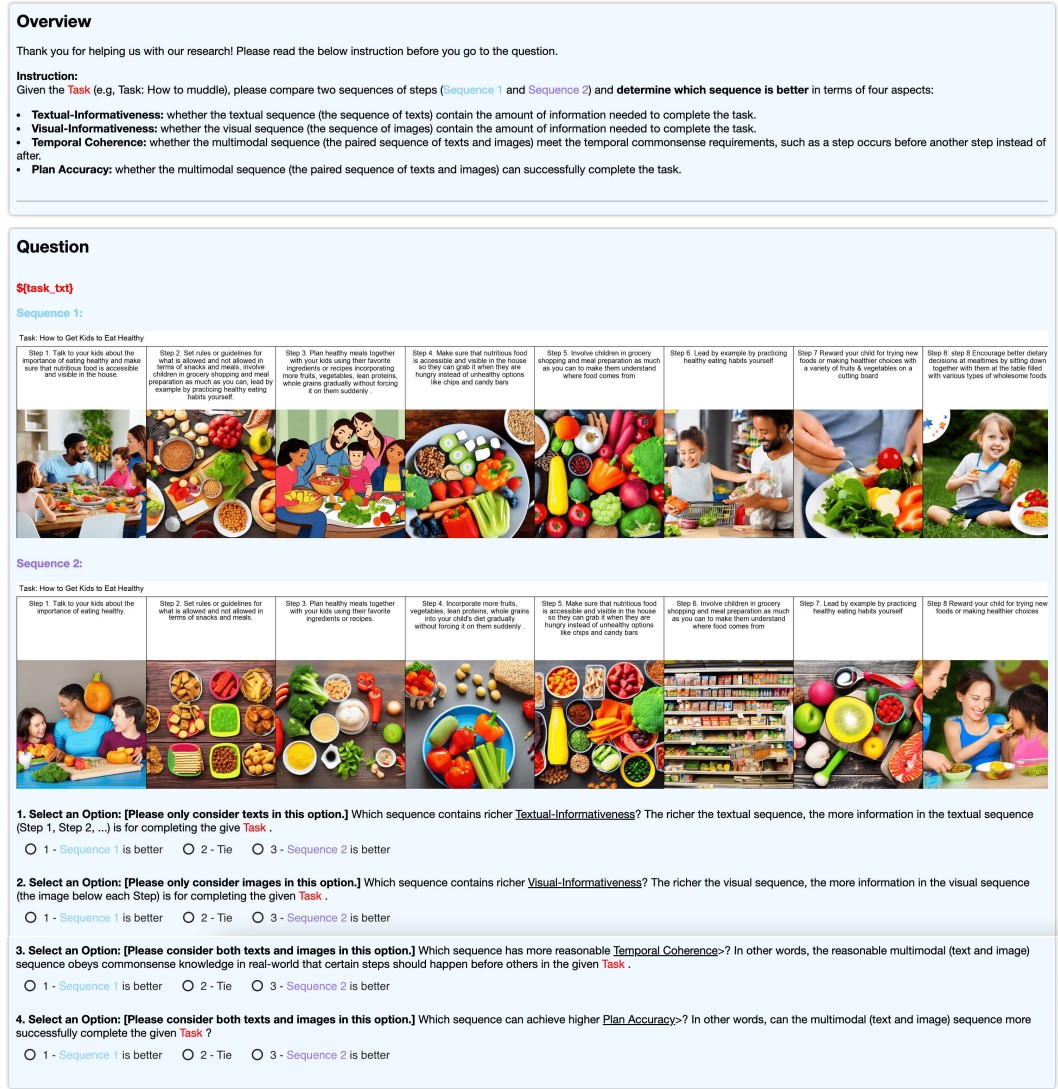

Figure 15: Amazon Mechanical Turk Platform. Questions Layout for Human Raters for Win-Tie-Lose Comparison on WIKIPLAN

## E.2 WORD CLOUD

In comparison with the word cloud distribution of the ground truth, we also show the word cloud of the baselines and TIP on WIKIPLAN and RECIPEPLAN.

## E.3 FAILURE CASES

In Figure 21, we showcase failure generation. For example, the state of the almond stays unchanged in Figure 21a, we suppose this is due to no explicit awareness of previous state change. In Figure 21b, at step 2, the generated image plan, though complemented with the text plan, loses authenticity in that the clock should not appear in a pan with carrots.

## E.4 LIMITATIONS

Relying on the LLMs to reason over complex text for text-to-image models though improving the quality, still remains a large gap with human performance. This is mainly restricted by the pre-

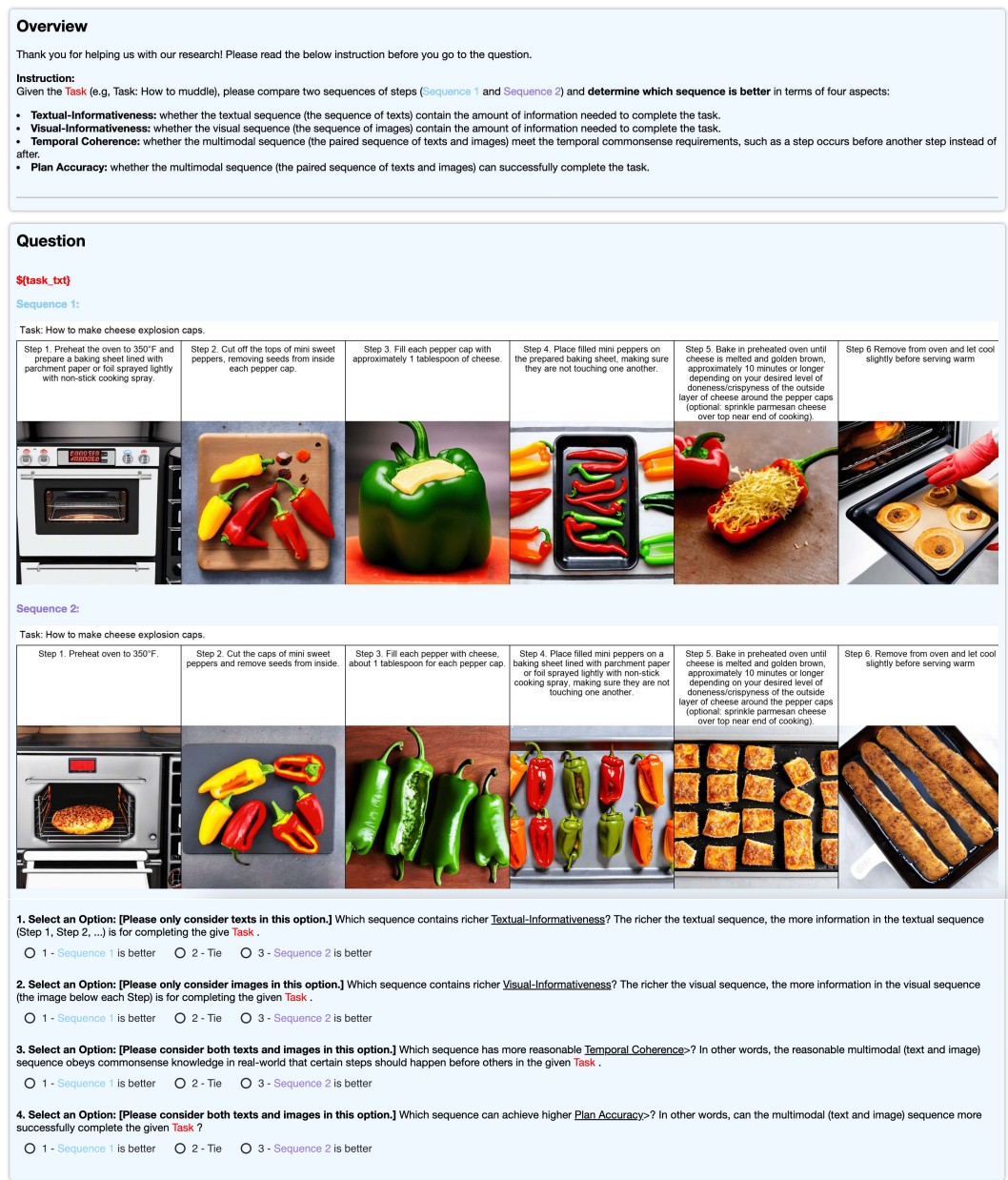

Figure 16: Amazon Mechanical Turk Platform. Questions Layout for Human Raters for Win-Tie-Lose Comparison on RECIPEPLAN.

training gap between LLMs and text-to-image models. To solve this, further work should explore the finetuning stage that how to inject this language reasoning into the multimodal generation models.

In addition to the model-side limitations, the dataset is limited in that not all the possible multimodal plans are provided and their quality is hard to validate. Due to the lack of perfect metrics in evaluating the text-image sequences, the research in multimodal procedural planning maybe difficult to scale up. Future work should explore this promising direction and furthermore lead LLMs and T2I models better multimodal procedural planners.

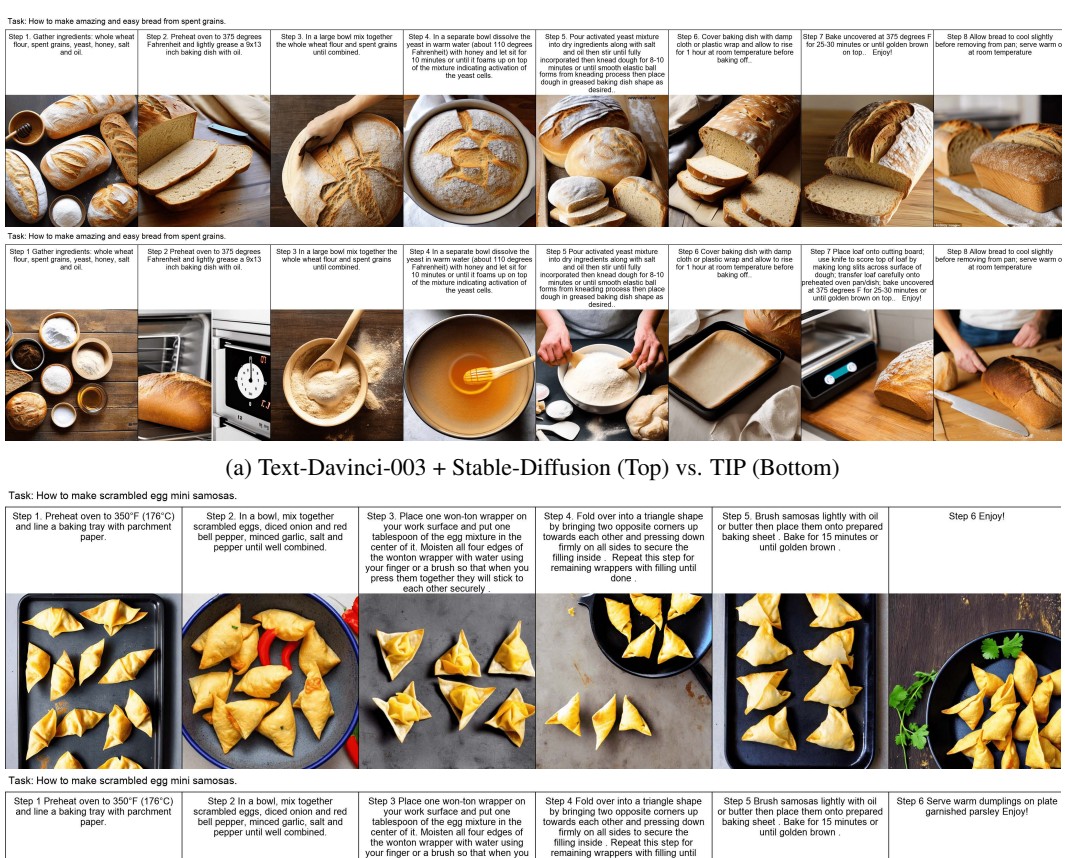

(a) Text-Davinci-003 + Stable-Diffusion (Top) vs. TIP (Bottom)

(b) Text-Davinci-003 + Stable-Diffusion (Top) vs. TIP (Bottom)

Figure 17: More showcases of plan comparisons on RECIPEPLAN.

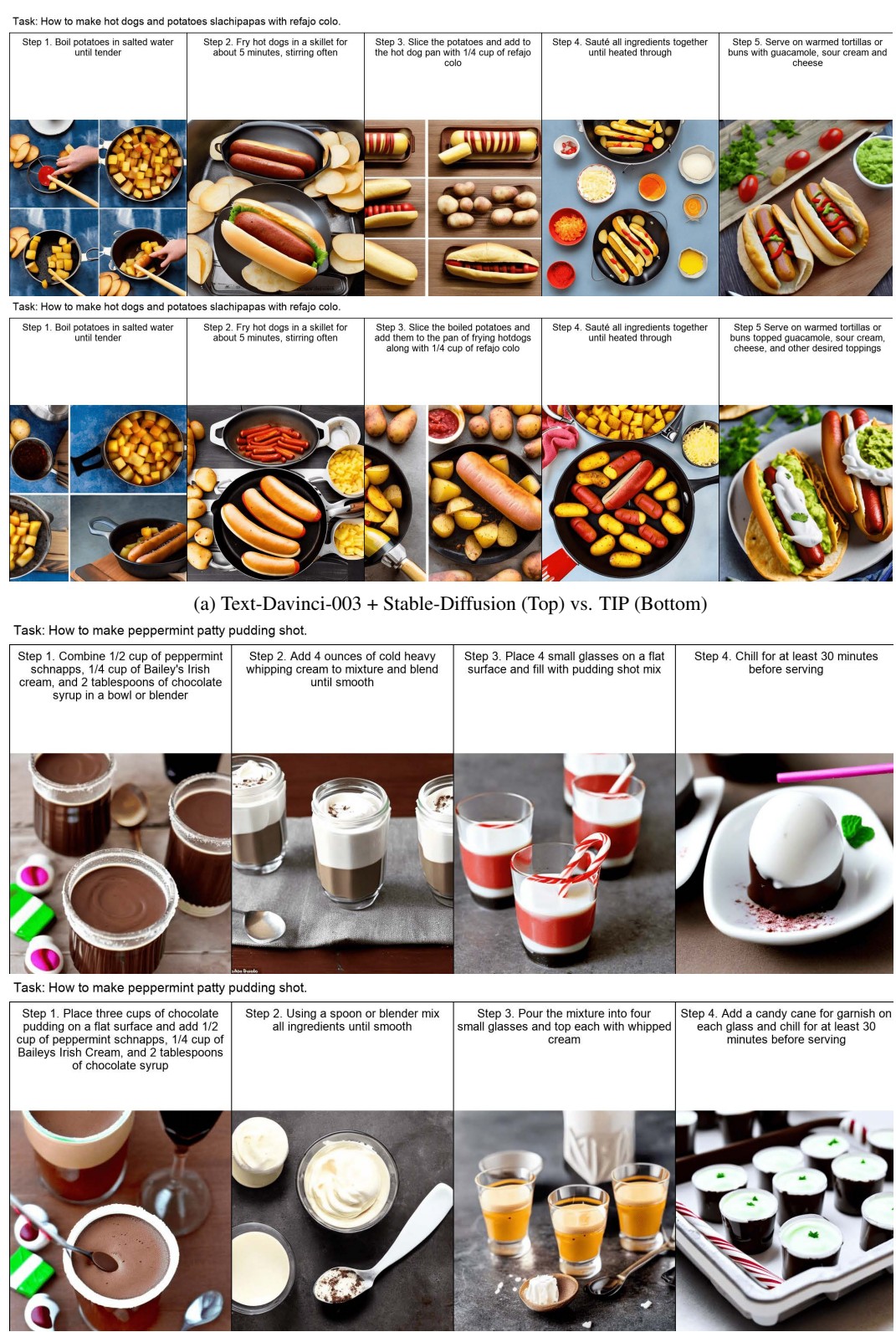

(a) Text-Davinci-003 + Stable-Diffusion (Top) vs. TIP (Bottom)

(b) Text-Davinci-003 + Stable-Diffusion (Top) vs. TIP (Bottom)

Figure 18: More showcases of plan comparisons on RECIPEPLAN.

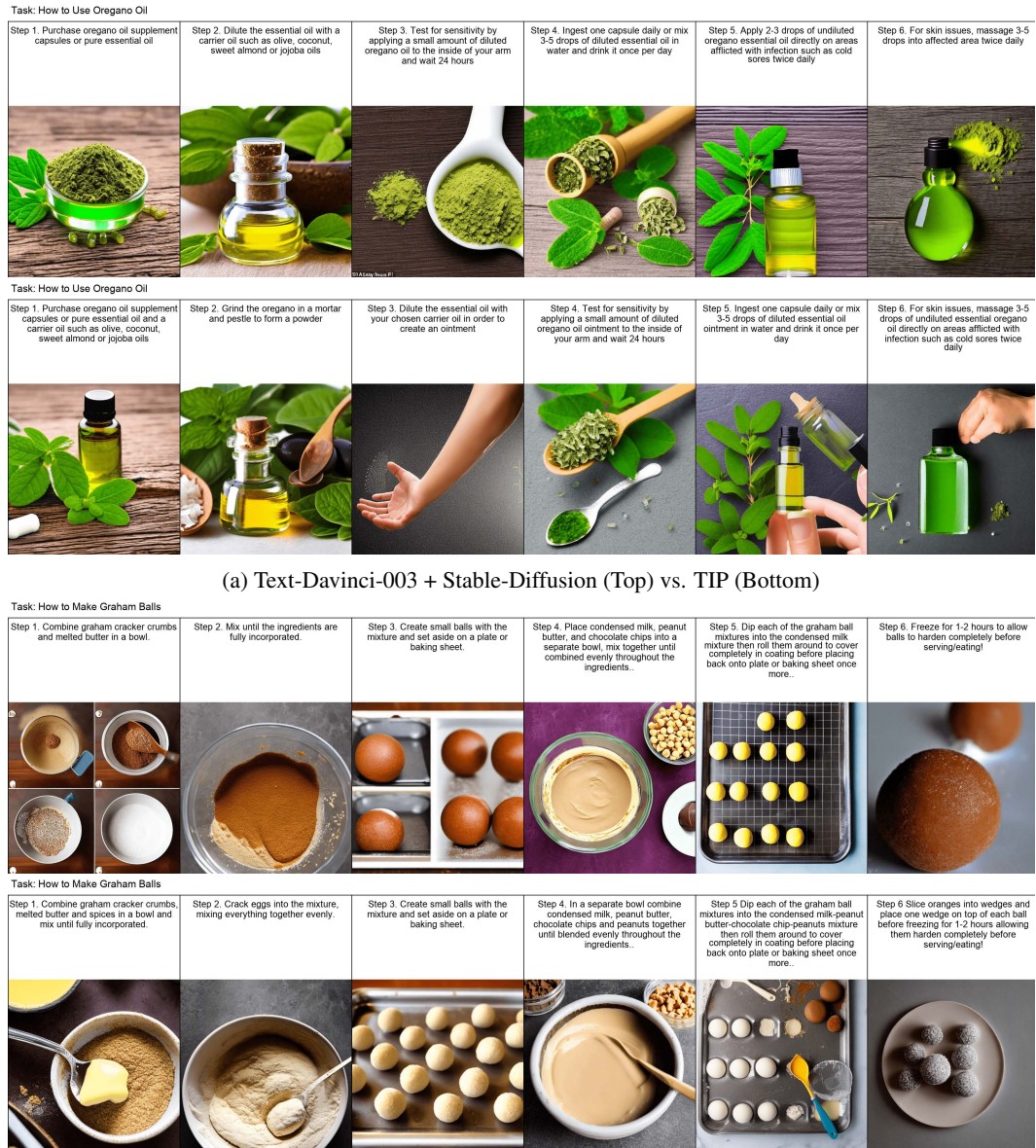

(a) Text-Davinci-003 + Stable-Diffusion (Top) vs. TIP (Bottom)

(b) Text-Davinci-003 + Stable-Diffusion (Top) vs. TIP (Bottom)

Figure 19: More showcases of plan comparisons on WIKIPLAN.

Task: How to Make Pizza Lunchables

| Step 1. Preheat oven to 400°F. | Step 2. Unwrap two Lunchables pepperoni pizzas and place them on a baking sheet lined with parchment paper or aluminum foil. | Step 3. Bake for 8 minutes, then remove from the oven and let cool for 5 minutes before serving. | Step 4. Cut each pizza into four slices using a pizza cutter or sharp knife, then serve with your favorite dipping sauce such as ranch dressing, barbecue sauce, ketchup, etc.. |

Task: How to Make Pizza Lunchables

| Step 1. Unwrap two Lunchables pepperoni pizzas and place them on a baking sheet lined with parchment paper or aluminum foil. | Step 2. Preheat oven to 400°F and bake for 8 minutes. | Step 3. Remove from the oven, let cool for 5 minutes before cutting each pizza into four slices using a pizza cutter or sharp knife. | Step 4. Place the slices in lunch box with chips and milk, then serve with your favorite dipping sauce such as ranch dressing, barbecue sauce, ketchup etc.. |

(a) Text-Davinci-003 + Stable-Diffusion (Top) vs. TIP (Bottom)

Task: How to Make Ice Balls

| Step 1. Fill an ice cube tray with water | Step 2. Place the tray in the freezer and freeze until solid | Step 3. Remove the frozen cubes from the tray | Step 4. Arrange three cubes together in a round shape and press them together lightly to form a ball shape | Step 5. Dip the ball into warm water for a few seconds to make it easier to handle | Step 6. Roll each ball on a flat surface or between two spoons until completely smooth and round |

Task: How to Make Ice Balls

| Step 1. Fill an ice cube tray with water and place it in the freezer until solid. | Step 2. Take out the frozen cubes from the tray and arrange them into a round shape, pressing lightly to form a ball shape. | Step 3. Dip the ball into warm water for a few seconds to make it easier to handle. | Step 4. Roll each ball on a flat surface or between two spoons until completely smooth and round, like rolling sugar cubes on a countertop or chocolate balls on a metal tray. | Step 5: a bowl of warm water, a person dipping an ice ball into the water, and an ice ball. | Step 6: a smooth, flat surface, like a cutting board, or two spoons, and a few ice balls being rolled between them. |

(b) Text-Davinci-003 + Stable-Diffusion (Top) vs. TIP (Bottom)

Figure 20: More showcases of plan comparisons on WIKIPLAN.

Task: How to make homemade raw almond butter.

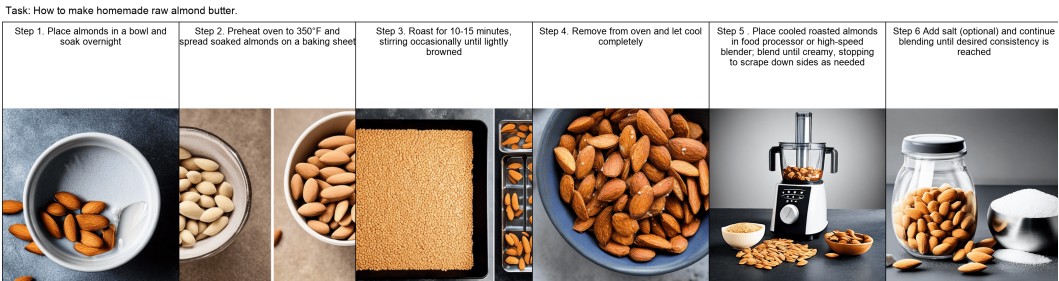

(a) Failure multimodal plans generated by our Text-Image Prompting (TIP).

Task: How to make carrot and swede potch mash.

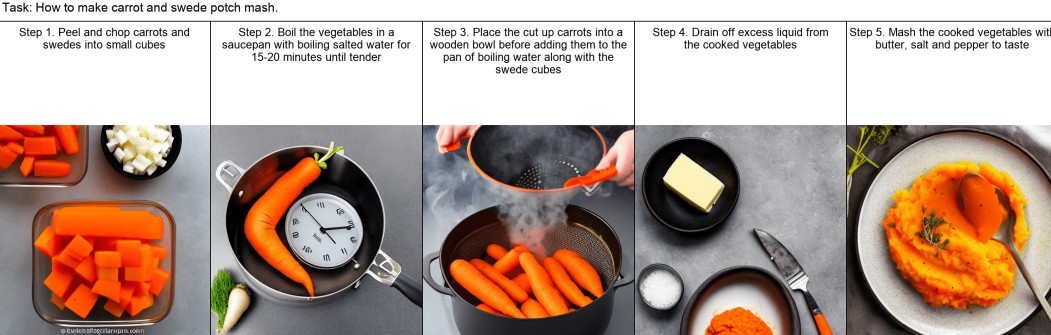

(b) Failure multimodal plans generated by our Text-Image Prompting (TIP).

Figure 21: We showcase failure cases of our Text-Image Prompting on generating multimodal plans on both datasets.

