# OpenReview forum: "Multimodal Procedural Planning via Dual Text-Image Prompting"
_ICLR.cc/2024/Conference — ICLR 2024 Conference Withdrawn Submission_

### Official Review · Reviewer_taPy · 2023-10-29

**Soundness:** 3 good
**Presentation:** 3 good
**Contribution:** 3 good
**Rating:** 5
**Confidence:** 4

**Summary:**

This article attempts to assist humans in completing tasks through the AIGC model. To do this, the authors first propose a multimodal program planning task, where the model can generate plans consisting of a series of image-text pairs to provide more information for achieving the goal. Secondly, to accomplish this task, the authors propose a method that combines LLMs and textual-visual graph models with text-image prompts. Finally, they collect two datasets, WIKIPLAN and RECIPEPLAN, to evaluate the proposed method.

**Strengths:**

1. A new task is proposed, with a high-level goal given, the model generates a plan composed of a series of image-text pairs.
2. I2T-Bridge and T2I-Bridge are proposed to improve the alignment of image-text pairs.
3. Two datasets were collected as benchmarks for the new task and were evaluated by human and automatic metrics.

**Weaknesses:**

1. In Table 2, the proposed TIP method mentioned in the text is not significantly better than the conventional Davinci-003 + SD in some metrics (S-BERT, ROUGE-L, METEOR). However, the win rate in Table 1 is around 60%. Why is that?

**Questions:**

1. What can I2T-Bridge specifically correct when it is used to correct textual instructions? If the images generated by the text-to-image model are not good enough, can the images be further corrected based on the corrected text?
2. How to solve the issue of object consistency in sequential images?
3. This article proposes a new task and presents a solution. What other problems remain unsolved, and how can these problems guide future research?

---

### Official Review · Reviewer_XpZ4 · 2023-10-30

**Soundness:** 2 fair
**Presentation:** 2 fair
**Contribution:** 3 good
**Rating:** 3
**Confidence:** 4

**Summary:**

This paper introduces the concept of multimodal procedural planning, which involves generating high-level goal-conditioned text and image plans to assist humans in completing tasks. The authors propose a method called Text-Image Prompting (TIP) to address the challenges of informativeness, temporal coherence, and accuracy of plans across modalities. TIP leverages LLMs and text-to-image generation models to jointly generate text and image plans. The authors collect two datasets, WIKIPLAN and RECIPEPLAN, to evaluate their approach.

**Strengths:**

- The paper introduces multimodal procedural planning, addressing an underexplored area, i.e., generating both text and image plans for high-level goals.
- Evaluation metrics w.r.t. generation quality are comprehensive, and the evaluation includes human ratings to assess the performance of the proposed approach.
- The work introduces two new datasets, WIKIPLAN and RECIPEPLAN, to evaluate the proposed task and approach.
- The authors also conduct a few robustness checks on the choice of templates used in T2I-B and I2T-B.

**Weaknesses:**

- The proposed method could improve the utilization of cross-modality. Cross-modal techniques such as soft prompting with embeddings or cross-modal attention mechanisms can make cross-modal matching more effective. Multimodal LLMs (Otter, MiniGPT, UniLM, GPT4, InstructBLIP, LLaVA, etc.) would also be more suitable for visual and language understanding.
- Although the paper provides a few baselines, the absence of baselines from existing text-based procedural planning works is a concern in assessing the usefulness of the proposed approach. Given that the paper makes use of Text Plan evaluation metrics, this part could be easily compared with recent procedural planning and script learning methods, e.g., [1,2].
- The proposed automated evaluation metrics access the generation quality. However, evaluation could be improved to assess 1) the degree of alignment or consistency between modalities, e.g., by using the Multimodal-Retrieval based metric proposed in [1], and 2) the quality of predictions or completions in these tasks, with evaluation tasks such as Next Step Prediction and Partial Sequence Completion from [2] or step reordering tasks.
- Even considering the current baselines, automated evaluation results in Table 2 appear marginal compared to the Text-Davinci-003 + Stable-Diffusion baseline that uses "LLMs and T2I models separately without collaboration". It is worth noting that marginal benefits are observed for the Multimodality Plan evaluation.
- Figure 4 examples do not appear very informative w.r.t. each of the subtasks. For example, part "read up on ocean safety tips and know the rules of the beach" is illustrated with a sign of text close to the beach reading "ocean safety" which does not relate to how safety signs would look in reality.
- It seems a bit difficult to extract valuable insights from the comparison in Fig. 6. Apart from perhaps slightly better images in Steps 4 and 8, the rest seem equally good to me.  Please consider adding more details to the highlighted parts, e.g., "Enjoy your freshly made juice!" is highlighted for TIP but appears on the output of both methods. Appendix E.1 contains several examples but is not accompanied by a description of what is observed (any insights) from these examples.

Overall the formulation of the proposed multimodal procedural planning task is of interest to the procedural planning and script learning communities, but the execution of the proposed method, baselines, and qualitative analysis require substantial revisions.

[1] Multimedia Generative Script Learning for Task Planning
[2] Non-Sequential Graph Script Induction via Multimedia Grounding

**Questions:**

- Can you further explain the results reported in Table 1? For example, first row, WikiPLAN, is TIP chosen as better from human evaluators 63.34% of the time over the Image Ref + OFA-Caption baseline?
- Can you also further explain the fact that, in terms of visual informativeness, the multimodal baselines can not achieve on-par results with textual reference-based baselines? Would it be that this is due to the lack of visual knowledge injected into LLMs, or perhaps a semantic gap between text and the generated images?
- What percentage approximately of the Imagined Prompts are object-centric and how many also involve actions or individuals?
- Can you share some statistics on the average step sequence/text length for the examples shown to human evaluators?

---

### Official Review · Reviewer_jid5 · 2023-11-01

**Soundness:** 2 fair
**Presentation:** 3 good
**Contribution:** 3 good
**Rating:** 6
**Confidence:** 3

**Summary:**

This paper introduces the Multimodal Procedural Planning task, which aims to generate text and image plans for completing tasks. The authors propose Text-Image Prompting (TIP), a dual-modality prompting method that combines large language models (LLMs) and text-to-image models to generate coherent and informative multimodal plans. They evaluate TIP on two benchmark datasets and show that it outperforms several baselines in terms of informativeness, temporal coherence, and plan accuracy.

**Strengths:**

1. The paper addresses an interesting and novel task of multimodal procedural planning.
2. The proposed TIP method is innovative and well-designed, leveraging the abilities of LLMs and text-to-image models to generate multimodal plans.
3. The paper conducts comprehensive evaluations and comparisons with baselines, demonstrating the effectiveness of TIP.

**Weaknesses:**

1. I'm curious about how image plan generation can enhance the vanilla text plan. Why does the Visual-Grounded Text Plan contain more information? Can you give some intuitive explanation? Also, do you any manual evaluations regarding the consistency between the generated images and the vanilla text plan?
2. Would your pipeline model introduce additional errors? What is the consistency rate between the Visual-Grounded Text Plan and the vanilla text plan? How do you prevent error accumulation in the pipeline model?

**Questions:**

Please check the weakness.